# Heatwaves reduce mating frequency in an aquatic insect

Md Tangigul Haque[1], Shatabdi Paul[1] and Md Kawsar Khan[1,2,3,*]

## ABSTRACT

Heatwaves are becoming more frequent and intense across the globe due to global warming. Heatwaves – unusual daytime and nighttime high temperatures over three consecutive days – can disrupt physiological functions of organisms, reducing fitness. Insects are stressed because of the increasing frequency and intensity of temperature extremes. While many studies have focused on insect behaviour during heatwaves in laboratory settings, the impact of natural heatwaves in the wild remains understudied. Here, we investigated the impact of natural heatwaves on mating behaviour, flight activity, and local abundance in the damselfly, *Xanthagrion erythroneurum*. We found that damselfly mating frequency decreased, while flight number and net population abundance remained unchanged during natural heatwaves. The decreased mating frequency may suggest a sex-specific decoupling of mate-searching efforts under thermal stress. Heatwave driven disruptions in mating behaviours and the occurrence of more frequent and acute heatwaves in the future may have long-term consequences for damselfly populations. Our results provide crucial data of the behaviour of thermally sensitive insects to heatwaves, which could assist in developing effective conservation strategies for maintaining biodiversity in a warming world.

KEY WORDS: Climate change, Extreme events, Conservation, Biodiversity, Insects

## INTRODUCTION

Climate change is increasing the frequency and intensity of extreme climate events – such as heatwaves – globally (Ragone et al., 2018; Tripathy et al., 2023). These rapid shifts in temperature can disrupt the biological functions of organisms and may contribute to biodiversity loss (Martinet et al., 2021). Ectotherms in particular are highly sensitive to heat stress due to their dependency on environmental temperatures for regulating physiological processes (Kingsolver et al., 2013). Thus, rapid and prolonged heat stress may push insects beyond their physiological limits, leading to changes in behaviour (Miler et al., 2020), physiology (González-Tokman et al., 2020), and morphology (Gerard et al., 2018) – especially affecting mating, flight activity, and population abundance. Understanding the impact of heatwaves in natural environments may provide valuable insights into

insects' resilience to extreme events, their population dynamics, and overall ecosystem stability.

In Australia, a heatwave is defined as a period of three or more consecutive days with higher maximum and minimum temperatures than a climate reference of that location (Nairn and Fawcett, 2013). The climate reference of a location at a particular time of the year is determined by the average temperature of past climatic data of that time of the year. The effects of heatwaves on insects are not homogenous and differ based on developmental stages, the timing of the stress events, the history of previous exposure, and the type of habitat (Carter et al., 2025; Murtaza et al., 2025). For example, exposure to heat stress at later developmental stages (closer to adult) in the diamondback moth (*Plutella xylostella*) resulted in lower reproductive performances than heat stress experienced at earlier life stages (Zhang et al., 2015). Extreme temperatures may also reduce in the number of individuals in open habitats compared to tree-covered or hedged habitats (Carter et al., 2025). A large proportion of studies that have investigated the impact of heatwaves on insect fitness have been largely studied in controlled laboratory conditions (Sales et al., 2018; Miler et al., 2020; Martinet et al., 2021; Li et al., 2023), mostly because of logistical challenges. While laboratory studies provide crucial data, the reduced natural complexity of such study designs may overlook or minimise the actual impact of heatwaves. Hence, studies in natural environment are required to accurately predict the impact of heatwaves on insects' fitness.

Odonata (dragonflies and damselflies) is a group of insects that acts as a bioindicator for assessing environmental condition in both freshwater and terrestrial ecosystems under global warming (Šigutová et al., 2019). Odonates are highly sensitive to both aquatic and terrestrial ambient temperatures, as their life cycle includes an aquatic larval stage and a terrestrial adult stage (Hassall, 2015). For example, a previous study found that higher temperature decreased egg development rates in four different species (*Celithemis elisa*, *Libellula luctuosa*, *Libellula pulchella*, and *Libellula incata*) of dragonflies in laboratory conditions (Frances et al., 2017). Responses to climate change that have so far been observed for damselflies include diapause, rapid larval development, and cyclical habitats use (Corbet, 1999; Šigutová et al., 2025). However, little is known about the responses of damselflies during natural heatwaves in the wild.

Insects respond to elevated temperatures by altering behaviours such as foraging, movement, and mating, which are important for their survival and reproduction (Sentis et al., 2015; Sánchez-Bayo and Wyckhuys, 2019; Miler et al., 2020). For example, ladybeetles (*Adalia bipunctata*) increase their egg-lying frequency in response to elevated temperature (Sentis et al., 2015), whereas grasshoppers show a reduced response to predator cues (Schmitz et al., 2016). These behaviour modifications can be either beneficial or maladaptive (Miler et al., 2020) and may have broader ecological consequences, resulting in changes in abundance and population dynamics. The consequences of heatwave events on insects' behaviour and life-history traits in a natural environment would provide robust data to determine the impact on population dynamics, yet such studies are uncommon. In this study, we investigated the impact of heatwaves on

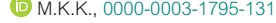

[1]School of Natural Sciences, Macquarie University, North Ryde 2109, Australia. [2]Department of Pharmacy, Chemistry and Biology, Freie University Berlin, 14195 Berlin, Germany. [3]Department of Applied Bioscience, Macquarie University, North Ryde 2109, Australia.

*Author for correspondence (bmbkawsar@gmail.com)

M.K.K., 0000-0003-1795-1315

mating behaviour, flight activity, and population abundance of a temperate damselfly *Xanthagrion erythroneurum* (Selys, 1876), predicting that heatwaves would reduce the number of mating pairs, flight activity, and abundance.

## RESULTS

The recorded temperatures during heatwave days ($n$=3) were 36.2°C, 37.3°C, and 39.5°C, while those on normal days ($n$=3) were 26.1°C, 30.9°C, and 28.2°C. Our sex-specific models showed no significant effect of heatwaves on flight number for either males ($N_{heatwaves}$=24, $N_{regular}$=36; glmmTMB: estimate=0.23±0.16, Z=2.48, P=0.13; Fig. 1A) or females ($N_{heatwaves}$=9, $N_{regular}$=6; glmmTMB: estimate=0.42±0.29, Z=1.44, P=0.15; Fig. 1B). Consistent with this, our additional model showed there was no significant interaction between days and sex (glmmTMB: estimate=−0.18±0.35, Z=−0.51, P=0.61), suggesting the flight number responses to heatwaves did not differ significantly between sexes. Similarly, we found no significant differences in damselfly abundance (glmmTMB: estimate=0.72±1.82, Z=0.39, P=0.69; Fig. 2A) between heatwave (mean=13.83±0.87 s.e.m.) and non-heatwave days (mean= 13.11±1.66 s.e.m.). Our data did provide evidence that heatwave days significantly reduced mating (glmmTMB: estimate=−2.16 ±0.76, Z=−2.82, P=0.004; Fig. 2B) (mean=0.38±0.18 s.e.m.) compared to non-heatwave days (mean=2.55±0.76 s.e.m.). We have summarised all our findings in Table 1.

## DISCUSSION

In this study, we showed that heatwaves in the wild reduced mating number in *X. erythroneurum* damselflies, without impacting flight number and overall abundance. Our study supports the prediction that heatwaves reduce mating in damselflies; however, we did not find evidence that heatwaves reduced flight number or abundance.

The reduced number of damselflies mating during heatwaves may arise from the physiological stress that they experience during excessive heat. High temperatures can alter male mating signals (Conrad, Stöcker and Ayasse, 2017), damage males' sperm production (Sales et al., 2018), impair male reproductive fitness (Sales et al., 2018), or reduce female receptivity to mating (Li et al., 2023). Recently, one study showed that experimental heatwaves can possibly damage mature sperm stored within the female's reproductive tract in flour beetles (*Tribolium castaneum*), halving subsequent fertility and negatively affecting offspring reproduction and lifespan (Sales et al., 2018). Our result aligns with previous studies that also found the negative consequences of heat stress on insect mating behaviour (Sales et al., 2018; Sales et al., 2021; Li et al., 2023). It is possible that in order to reduce thermal stress, females prioritised other adaptive activities by seeking shaded areas over mating during heatwaves, as mating requires more energy, reduces foraging and increases predation risk.

We did not find a significant impact of heatwaves on the flight number of male and female damselflies. The animals may have been able to maintain flight even on very hot days via effective thermoregulatory behaviours or physiological adaptations such as reduced basking behaviour and wing positioning (Sheikh and Douglas, 2012). Similar to our finding, one study reported that the flight activity levels of damselflies (*Enallagma doubledayi*) were not associated with changes in ambient temperature (Mason, 2017). The unexpected high flight number on hot days may be associated with searching for suitable mates, patrolling territories, foraging, seeking cooler microhabitats, or dissipating heat (May, 1995; Dudley, 2002; Carter et al., 2025). For example, insects such as bumble bees can optimise their muscle performance via regulating thorax temperature and so prevent overheating during flight (Dudley, 2002). In contrast, temperatures above species optima level or upper thermal limits (critical thermal maxima, CTmax) significantly reduce species' flight activity (Glass and Harrison, 2022).

Our results demonstrated that heatwaves did not significantly reduce damselfly abundance near the pond. Many species deal with heatwaves by changing their microhabitat – such as to an area with greater shade – to reduce thermal stress. For example, moose (*Alces alces*) increase visits to shaded areas and water sources during heatwave days (Alston et al., 2020). Damselflies exhibit microhabitat variation across developmental stages (Khan and Herberstein, 2020) and could have similarly switched from the pond to a more shaded forest area in order to reduce thermal stress; however, we did not find evidence in support of this behaviour. Instead of moving to a more shaded forest area, damselflies most likely utilised pond vegetation for shade and thereby reduced thermal stress without losing mating and oviposition opportunities. We have observed that damselflies move to shaded areas or lower parts of plants and stay near water bodies, particularly during heatwave conditions (systematic data was not collected and analysed). By actively moving to cooler microhabitats, damselflies may reduce the risk of overheating, heat-related physiological stress, and mortality by avoiding raising their body temperature above critical thermal limits. This short term buffer against temperature extremes may contribute to maintain abundances

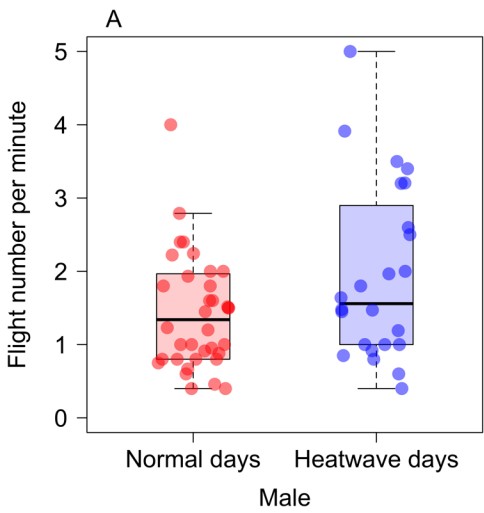

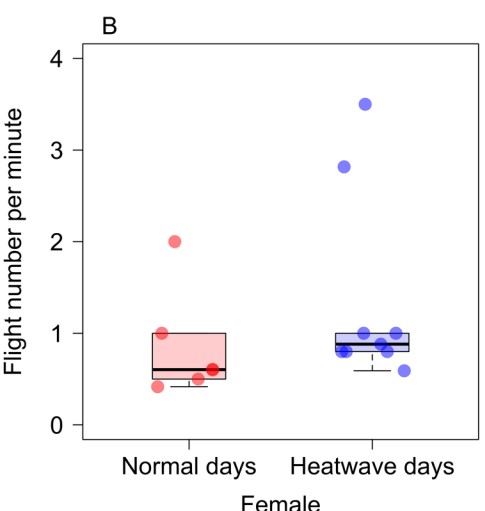

**Fig. 1. Differences in damselfly flight number per minute during heatwave and normal days.** Boxplots showing no significant differences between (A) male and (B) female damselfly flight numbers during heatwave and normal days. Bold line inside boxes denotes median, the bottom and top lines of the boxes represent 25th and 75th percentiles, respectively. Blue and red dots depict flight number per minute for an individual during heatwaves and normal days, respectively. Male sample size ($N_{heatwaves}$=24, $N_{regular}$=36); female sample size ($N_{heatwaves}$=9, $N_{regular}$=6).

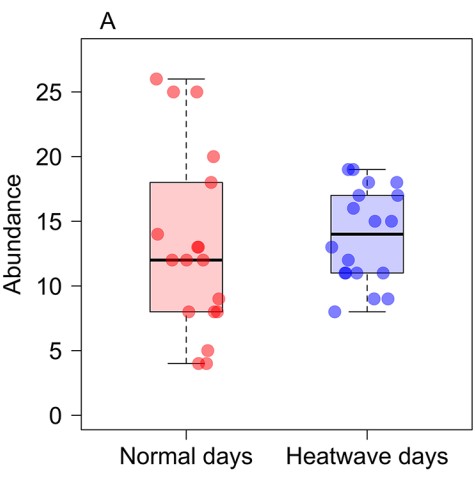

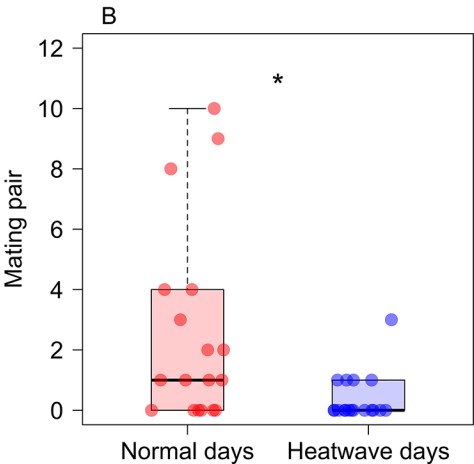

**Fig. 2. Differences in abundance and mating pairs during heatwave and normal days.** The boxplots show that (A) abundance of damselflies was similar during heatwave and normal days, and (B) the number of mating pairs was lower during heatwave days. Bold line inside boxes denotes median, the bottom and top lines of the boxes represent 25th and 75th percentiles, respectively. Each red and blue dot represents a data point collected during normal days and heatwave, respectively. We used 36 observations to determine abundance (*N*=485) and 36 observations to count mating pairs (*N*=53). *$P<0.05$.

at normal levels even at elevated temperatures. The quantification and significance of such behaviours in regulating thermal stress is a future research avenue towards understand heatwave resilience in insects. Additionally, insects with high reproductive rates and short generation times can quickly recover from population loss due to heat stress. For example, male flour beetles (*T. castaneum*) can recover from heat stress and return to regular fertility after 15-28 days post exposure (Sales et al., 2021). Nevertheless, it is alarming that many studies have projected an increasing frequency of heatwaves and a rise in global temperatures in the future, which may have a larger impact on insect abundance (Perkins-Kirkpatrick and Lewis, 2020; Tripathy et al., 2023) than the results we have reported in this paper.

Overall, our study showed that heatwave events reduce mating number but do not significantly affect damselfly flight number and abundance. The responses we observed due to heatwaves have a significant fitness cost and could contribute to the decline of damselflies, which are already under stress because of the impact of other global change factors such as pollution, microplastics and pharmaceuticals (Wagner et al., 2021; Khan and Rolff, 2025). While direct mortality and fitness costs on physiological traits such as body size are well studied, indirect fitness costs – especially in mating and other reproductive traits such as sperm quality, viability and fecundity – remain poorly understood, albeit a significant contributor of population decline. Our study provides key evidence that heatwaves reduce mating in damselflies, which could contribute to their population decline. The extent of mating reduction and other fitness costs of heatwave stress would vary depending on heatwave duration and frequency. Though our study provides evidence of the impact of heatwaves on insects within an ecological context in natural conditions, there are a few complexities in performing

observations in the field. It is challenging to identify the sole impact of heatwaves on insects due to the influence of other environmental factors such as food availability, humidity, and wind speed. Additionally, logistical constraints resulted in a limited sample size. However, it is necessary to perform field-based observations to understand the actual impact of natural heatwaves on insects in the wild in order to accurately predict insect resilience to climate change. Australian Bureau of Meteorology data from the last five years (2020-2024) indicate that damselflies are frequently experiencing acute temperatures (average number heatwave days per year=7 days), and the trend will continue, which may contribute to population and species decline in the future (Cowan et al., 2014; Jyoteeshkumar reddy et al., 2021). Our study, by providing crucial evidence of the impact of heatwaves on wild populations, helps to better understand species response to extreme climatic events such as heatwaves, and highlights the importance of species conservation and biodiversity management in the face of climate change.

## MATERIALS AND METHODS

We collected data on damselfly activities (e.g. mating behaviour, flight number, and abundance) at Macquarie University pond (33.772 S, 151.114 E) in North Ryde, Australia, during natural heatwaves and normal (non-heatwave) periods in December 2024, and January and February 2025. In this study, we identified a heatwave as a period of three consecutive days or more when the average temperature substantially exceeded the historical monthly average temperature for the respective month, based on 1859-2020 climate records. The historical monthly average temperatures for December and January were 21.5°C and 22.4°C, respectively, for our study site. We identified three heatwave events where the temperature exceeded the historical monthly average by at least 4°C: 15-17 December 2024 (26.7°C, +5.2°C above average), 25-27 December 2024 (25.5°C, +4°C above average), and 26-28 January 2025 (27.9°C, +5.5°C above average). Using

**Table 1. Summary of model outputs for each response variable**

| Model | | Response variable | Fixed factor | Random factor | Output | | | |
|---|---|---|---|---|---|---|---|---|
| | | | | | Estimate | Standard error | Z value | *P*-value |
| Model 1 | Male | Flight number | Days | Date | 0.23 | 0.16 | 2.48 | 0.13 |
| | Female | Flight number | Days | Date | 0.42 | 0.29 | 1.44 | 0.15 |
| Additional model (male+female) | | Flight number | Days×sex | Date | −0.18 | 0.35 | −0.51 | 0.61 |
| Model 2 (male+female) | | Abundance | Days | Study site+date | 0.72 | 1.82 | 0.39 | 0.69 |
| Model 3 (male+female) | | Mating number | Days | Study site+date | −2.16 | 0.76 | −2.82 | 0.004 |

Note: 'Days×sex' refers to the interaction between days (heatwave and normal days) and sex (male and female).

the prediction of the Bureau of Meteorology, we identified heat peak of heatwave days and collected behavioural data during 2-3 h of peak heat. We recorded field site temperature during the heat peak using the wind meter (anemometer, Kestrel-3000; USA). We collected behavioural data on three separate days for each condition. Data for normal days were collected an average of 9.6±1.5 s.e.m. days after the heatwave sampling periods. Three days of heatwave data and historical data were collected from the Bureau of Meteorology, Australia.

## Flight activity

Flight number was monitored as a proxy for damselflies' foraging behaviour, activity levels and energy expenditure. We captured each damselfly using a sweep net (Paul et al., 2024; Haque et al., 2025) and marked them on the wing using a marker pen to track their activity. After marking, we placed the damselfly back into its habitat (on a small branch of a plant) and followed their activity after ~10 s of acclimation period. We used gloves to minimise any potential factors that could influence damselfly performance. We recorded data on the number of flights by an individual when we were able to observe a marked individual for a minimum of 2 min, up to a maximum of 5 min. We spent 45 min recording flight numbers of damselflies from three sites (15 min in each site) and pooled the data from all sites for each day for analysis.

## Mating behaviour

We collected data on mating pairs for 5 min at each site (three sites) during heatwave (3 days) and non-heatwave (3 days) conditions. We recorded a mating pair when we observed a male and female in a mating tandem or a mating wheel. After counting each pair, we separated the mating pair and marked them to avoid counting the same pair multiple times. We did not exclude marked individuals that had previously been observed for flight activity, as mating activity and flight activity were recorded separately at each site. While we separated and marked mating pairs to avoid multiple counting, there was a minimal possibility that separated individuals may have rejoined or formed pair with the same individual or other individuals. To avoid counting the same pair, we did not count them if both male and female were marked but included a pair if they formed new pair, indicated by one marked individual and one unmarked individual.

## Abundance

We measured damselfly abundance for 5 min from three sites using a capture-and-release method (Khan, 2020). We walked through the sites and captured damselflies with an insect sweep net and marked their wings to avoid multiple counting. We counted total number of damselflies on each site to estimate their abundance.

## Statistical analyses

We applied generalised linear mixed models using template model builder (glmmTMB) to assess the impact of heatwaves on damselfly flight activity, mating behaviour, and local abundance (Brooks et al., 2017). We fitted models using flight number as the response variable and days (heatwave and normal days) as a fixed factor, and date as a random factor using sex-specifics subsets of the data to observe the effect of heatwaves on flight number within each sex independently. We also added an additional model using the full dataset, including sex and days as fixed factors, flight number as the response variable, and study date as a random factor, to observe whether the effect of heatwaves differed between sexes. To determine heatwave impact on mating and abundance, we fitted models with mating pair abundance as a response variable and days (heatwave and normal days) as fixed factor, and site and date as a random factor. We used DHARMa package in R to check the residual plots and goodness-of-fits of our models (Hartig, 2016). We performed all analyses in R (version 4.1.2) (R Core Team, 2021) using packages glmmTMB (Brooks et al., 2017) and Dharma (Hartig, 2016).

## Acknowledgements
We acknowledge the traditional custodians of the land the Wallumattagal clan of Dharug nation where Macquarie University is situated, and where we conducted our research. We also acknowledge the support provided by the Behavioural Ecology Lab at Macquarie University, including resources that contributed to the completion of this study.

## Competing interests
The authors declare no competing or financial interests.

## Author contributions
Conceptualization: T.H., S.P., K.K.; Data curation: T.H., S.P.; Formal analysis: T.H., S.P.; Investigation: K.K.; Methodology: T.H., S.P., K.K.; Resources: K.K.; Supervision: K.K.; Visualization: T.H.; Writing – original draft: T.H., S.P.; Writing – review & editing: K.K.

## Funding
No funding was required for conducting the research. Open Access funding provided by Macquarie University. Deposited in PMC for immediate release.

## Peer review history
The peer review history is available online at https://journals.biologists.com/bio/lookup/doi/10.1242/bio.062091.reviewer-comments.pdf

## Data and resource availability
All relevant data and code used in this study are available on Figshare and can be accessed via following link: https://figshare.com/s/bb6799372395367b7ca9.

## First Person
This article has an associated First Person interview with the first author of the paper.

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
