## [Peer Review File · Biology Open]

Heatwaves reduce mating frequency in an aquatic insect

Md Tangigul Haque, Shatabdi Paul and Kawsar Khan

DOI: 10.1242/bio.062091

Editor: Kendra J. Greenlee

Review timeline

Original submission:	5 May 2025
Editorial decision:	13 May 2025
Resubmission:	28 May 2025
Editorial decision:	9 June 2025
First revision received:	1 July 2025
Accepted:	7 July 2025

Original submission

First decision letter

MS ID#: bio.062053

MS Title: Heatwaves reduce mating frequency in an aquatic insect

Authors: Md Tangigul Haque; Shatabdi Paul; kawsar khan

Dear Dr Khan,

I am writing to let you know that I have now reached a decision on the above manuscript.

As you will see, the reviewers raised a number of substantial criticisms that prevent me from accepting your paper for publication.

The reviewer reports are shown at the bottom of this email or can be accessed, together with a copy of this decision letter, by going to:

Although both reviewers agreed that this topic is important, they expressed concerns about the methods and statistical analysis. I do hope you find the reviewer comments helpful in allowing you to revise the manuscript for successful submission to Biology Open or elsewhere.

Reviewer 1

Comments for the author

In the present manuscript, Tangigul Haque and colleagues present a study investigating the impact of heatwaves on the reproductive biology of the damselfly, *Xanthagrion erythroneurum*. Specifically, the study compares flight activity, abundance, and the number of mating pairs during natural heatwaves and normal periods across three pond sites near North Ryde, Australia. Given that climate change is expected to increase the frequency and intensity of heatwaves, understanding how reproductive behavior in ectotherms responds to such conditions is indeed a relevant and timely topic in conservation biology. However, there are significant flaws in the experimental design and data analysis that, in my view, compromise the validity of the conclusions drawn.

My major concerns are as follows:

1) It is unclear over how many days the behavioral observations were recorded. Were different days treated as biological replicates under both normal and heatwave conditions? Unfortunately, the manuscript lacks a supplementary table or section clearly detailing this information.

2) In line 111, the authors present average temperatures for heatwave and normal days. However, the source of this data is not specified. Was this based on local meteorological records, or do the authors have microclimatic data (temperature and humidity) collected directly from the study sites?

3) The rationale for not using a standard null hypothesis significance testing approach (e.g., GLM) is not provided. Instead, the authors rely on the DurgaDiff function, which is an effect size analysis. While such approach can be valuable in some circumstances, the authors do not explain this choice adequately. Moreover, in several comparisons (e.g., Figure 1A, 1B, and 2A), the confidence intervals include zero, indicating that the observed differences are not statistically meaningful.

In summary, while the research question is relevant and aligned with current conservation concerns, the manuscript presents limitations in the experimental design and statistical analysis at this stage.

Reviewer 2

Comments for the author

This study used an observational approach in natural ecosystems to study the impacts of heatwaves on damselfly behavior and mating success. The authors tracked flight activity, mating pairs and abundance of the damselfly *Xanthagrion erythroneurum* at three pond sites during natural heatwave days ($\geq 35^\circ\text{C}$) versus normal summer days. They find that heatwaves reduced mating frequency, slightly boosted male flight activity, and did not affect local abundance, indicating that brief extreme temperatures can decouple mate-searching from successful reproduction without immediately thinning populations.

I think the study provides a valuable approach, focused on observations in natural environments, to study the effects of heat waves, and has the potential to be a useful contribution. However, in its current form I think there are a couple of important issues that would need to be addressed before the manuscript could be considered for publication.

First, the details of the statistical approaches used are not sufficient to assess the validity of the key findings. In particular, details on inclusion of covariates associated with environmental conditions or sampling regime need to be addressed, as well as specifying in more detail the statistical tests used.

Second, the treatment of weather data and definitions of heat waves need to be further explained. For example, the study defines 'heat wave' days as those above 35°C , but it is not specified whether this is the peak heat, a duration of time exceeding that threshold, etc. It is also not clear from the Methods section what data were used to make this determination, for example whether data were taken from a local weather station, meteorological data, etc. In addition to concerns about the timing of sampling, these microclimatic variations might be important for interpreting the environmental conditions actually experienced by the damselflies.

Finally, a suggestion - if weather data for the specific timing of observation events is available - would be to analyze the effects of temperature as a continuous variable (e.g., temperature during observation period), rather than a discrete variable (heat wave days vs. normal).

Comments on these and other specific aspects of the manuscript are included below:

Specific comments

L47: add 'and' after '(6),'

L49: incorrect possessive on insects: should be 'insects'

L52: 'defined as the unusual high daytime' should be edited to 'defined as unusually high daytime'. Also, a more specific definition should be used here - what formal definition of heat waves are the authors using? What defines 'unusually high'?

L61-63: I agree on the importance of study insects' responses to e.g., stress in naturally complex conditions. But would also be good to highlight some of the potential limitations of this approach (either here or in the discussion)

L66: 'bioindicators' should be singular

L67: remove 'the'

L68-69: sentence needs to be reworded. maybe 'to both aquatic and terrestrial ambient temperatures'?

L77:78: I think this is a good point and worth expanding. I would suggest emphasizing behavior (and providing a bit more background on the impacts of heat on insect behavior), given the focus on behavior in the current study

L84-87: More details on the sampling approach are needed here. Were heat wave days defined ahead of time and targeted for sampling? Or was sampling random, and then heat wave days defined post-hoc by environmental temperature data? Related, how was the heat wave for the day defined? A single peak temperature reading above 35C? Or some amount of time spent above that threshold? This is important because temperatures will fluctuate significantly throughout the day, meaning the timing of observations would significantly affect body temperatures experienced. And a final important point here is that the time window of when sampling occurred is not defined - does this occur during a standardized time period? Likewise, where are these weather data gathered from? Meteorological data? Or a local weather station?

L96: confusing wording here: should maybe be '... minimum of 2 minutes. Individuals were observed for up to a maximum of 5 mins.'

L96-98: Is the 15 mins per site, or per individual? Were multiple individuals recorded at each site?

L100-101: More detail would be helpful here - were the marked individuals from above included in these mating pairs? Were they only counted, or were other observations recorded? And is there a possibility that mating pairs rejoined, and if so would they potentially be recounted?

L106-108: The details of the statistical methods here should be significantly expanded. What specific statistical tests are implemented in this package? Related, how were the observations distributed across days and sites? If multiple observations occurred at the same sites or on the same days, were these factors incorporated in statistical models (e.g., as random effects in a mixed model)?

L111-112: Are the maximum temperatures? Or averages?

L113: statistical tests used and test statistics should be reported here and throughout. For example, how were these confidence intervals generated? In addition to potentially conflated factors (e.g., spatial or temporal replication), it appears that at least some of the variable (e.g., flight #) are not normally distributed, so it is unclear whether, for example, confidence intervals generated assuming a normal distribution would be appropriate here.

L126: remove 'noticeable'

L138: 'insects' should be singular

L152: Change 'denoted' to 'demonstrated' or equivalent?

L159-162: interested observation, I would suggest expanding on the details of this observation a bit.

L162: '... and may also contributed' needs to be reworded

L175: 'significance' should be 'significant'

L185: Word missing in this sentence, maybe 'are experiencing'?

Fig 1: I would suggest either plotting kernel density estimates or boxplots, but not both - I find this distracting (especially given that the raw data is also plotted)

Reviewer's Responses to Questions

Experimental quality

Does each figure have the proper controls?

If 'No', please indicate reasons in Comments for Author box below.

Reviewer #1:

- Yes

Reviewer #2:

- Yes

Were the data analyzed using appropriate statistical tests?

If 'No', please indicate reasons in Comments for Author box below.

Reviewer #1:

- No

Reviewer #2:

- No

Reproducibility

Were experiments performed using adequate number of biological replicates?

If 'No', please indicate reasons in Comments for Author box below.

Reviewer #1:

- No

Reviewer #2:

- No

Does the methods section provide sufficient detail to permit reproducibility?

If 'No', please indicate reasons in Comments for Author box below.

Reviewer #1:

- No

Reviewer #2:

- No

Completeness

Are the manuscript's conclusions supported by the data?

If 'No', please indicate reasons in Comments for Author box below.

Reviewer #1:

- No

Reviewer #2:

- No

Scholarship

Do the authors cite and discuss the merits of data that would argue for and against their conclusion?

If 'No', please indicate reasons in Comments for Author box below.

Reviewer #1:

- Yes

Reviewer #2:

- Yes

Does the manuscript title & abstract accurately reflect the contents of the manuscript, without hyperbole?

If 'No', please indicate reasons in Comments for Author box below.

Reviewer #1:

- No

Reviewer #2:

- Yes

Author response to reviewers' comments

Dear editor,

Thank you for your valuable suggestions. We addressed all the comments that reviewers suggested to improve the quality of our article. We incorporated all changes in our main article and added here in-details with line numbers for your convenience. Please find our responses below:

Reviewer 1:

It is unclear over how many days the behavioral observations were recorded. Were different days treated as biological replicates under both normal and heatwave conditions? Unfortunately, the manuscript lacks a supplementary table or section clearly detailing this information.

Response: We now clarified the requested in line 98.

In line 111, the authors present average temperatures for heatwave and normal days. However, the source of this data is not specified. Was this based on local meteorological records, or do the

authors have microclimatic data (temperature and humidity) collected directly from the study sites?

Response: We collected data directly from field site using Anemometer. We have added this information in line 96-98.

The rationale for not using a standard null hypothesis significance testing approach (e.g., GLM) is not provided. Instead, the authors rely on the DurgaDiff function, which is an effect size analysis. While such approach can be valuable in some circumstances, the authors do not explain this choice adequately. Moreover, in several comparisons (e.g., Figure 1A, 1B, and 2A), the confidence intervals include zero, indicating that the observed differences are not statistically meaningful.

Response: We now applied generalized linear mixed models (GLMM) for determining statistical significance. Effect size were calculated using DurgaDiff function by bootstrapping data 1000 times, DurgaDiff does not require normal distribution of data and ideal for determining effect size and confidence interval. In have added details information in method (line 131-141) and in result (line 146-158).

Reviewer 2:

First, the details of the statistical approaches used are not sufficient to assess the validity of the key findings. In particular, details on inclusion of covariates associated with environmental conditions or sampling regime need to be addressed, as well as specifying in more detailed the statistical tests used.

Response: We applied generalized linear mixed models (GLMM) as per suggestion of the reviewer with environmental conditions such as collection days and sampling sites as random factor in our analysis. We provided information in method (line 131-141), and in result (line 146-158).

Second, the treatment of weather data and definitions of heat waves need to be further explained. For example, the study defines 'heat wave' days as those above 35C, but it is not specified whether this is the peak heat, a duration of time exceeding that threshold, etc. It is also not clear from the Methods section what data were used to make this determination, for example whether data were taken from a local weather station, meteorological data, etc. In addition to concerns about the timing of sampling, these microclimatic variations might be important for interpreting the environmental conditions actually experienced by the damselflies.

Response: We collected weather data from using a weather data recorder (Anemometer). We have clarified how we described heatwave in line 52-53, and added information in line 93-98.

Finally, a suggestion - if weather data for the specific timing of observation events is available - would be to analyze the effects of temperature as a continuous variable (e.g., temperature during observation period), rather than a discrete variable (heat wave days vs. normal).

Response: Our aim of this study was to determine how thermoregulatory behaviour changes during heatwaves. We designed this study to collect data from field during heatwave and normal days. We agree with the reviewer that using temperature as continuous variable would provide information how specific degree of temperature changes would alter behaviour. We, however, did not do that for two reason: 1) that would be hypothesis generation of post data collection and 2) our sample number will be limited to provide strong evidence. We would consider in future studies to collect data at different temperature to test temperature change effect on thermoregulatory behaviour.

Specific comments

L47: add 'and' after '(6),'

Response: We have added the information in line 47.

L49: incorrect possessive on insects: should be 'insects'

Response: we have corrected the word in line 49.

L52: 'defined as the unusual high daytime' should be edited to 'defined as unusually high daytime'. Also, a more specific definition should be used here - what formal definition of heat waves are the authors using? What defines 'unusually high'?

Response: We have re-written the definition here in line 52-53. Also, we used more specific definition in our study that we mention in line 93-94.

L61-63: I agree on the importance of study insects' responses to e.g., stress in naturally complex conditions. But would also be good to highlight some of the potential limitations of this approach (either here or in the discussion)

Response: We have added limitations in discussion section in line 222-229.

L66: 'bioindicators' should be singular

Response: We have corrected the form in line 66.

L67: remove 'the'

Response: We removed the information in line 67.

L68-69: sentence needs to be reworded. maybe 'to both aquatic and terrestrial ambient temperatures?'

Response: We have reworded the sentence in line 68-69.

L77:78: I think this is a good point and worth expanding. I would suggest emphasizing behavior (and providing a bit more background on the impacts of heat on insect behavior), given the focus on behavior in the current study

Response: We have expanded the information in line 77-82.

L84-87: More details on the sampling approach are needed here. Were heat wave days defined ahead of time and targeted for sampling? Or was sampling random, and then heat wave days defined post-hoc by environmental temperature data? Related, how was the heat wave for the day defined? A single peak temperature reading above 35C? Or some amount of time spent above that threshold? This is important because temperatures will fluctuate significantly throughout the day, meaning the timing of observations would significantly affect body temperatures experienced. And a final important point here is that the time window of when sampling occurred is not defined - does this occur during a standardized time period? Likewise, where are these weather data gathered from? Meteorological data? Or a local weather station?

Response: We define heatwave days ahead of time and targeted for sampling. We have now clarified sampling event and provided further data to clarify how data fulfill definition of heat wave and added all details in line 95-98.

L96: confusing wording here: should maybe be '... minimum of 2 minutes. Individuals were observed for up to a maximum of 5 mins.'

Response: We have corrected the sentence in line 108.

L96-98: Is the 15 mins per site, or per individual? Were multiple individuals recorded at each site?

Response: 15 mins per site, we clarified in line 109-110.

L100-101: More detail would be helpful here - were the marked individuals from above included in these mating pairs? Were they only counted, or were other observations recorded? And is there a possibility that mating pairs rejoined, and if so would they potentially be recounted?

Response: We have added more details in line 116-122.

L106-108: The details of the statistical methods here should be significantly expanded. What specific statistical tests are implemented in this package? Related, how were the observations distributed across days and sites? If multiple observations occurred at the same sites or on the same days, were these factors incorporated in statistical models (e.g., as random effects in a mixed model)?

Response: We applied generalized linear mixed models (GLMM) as per suggestion of the reviewer using site and date as random factor. We have added details information in line 131- 141.

L111-112: Are the maximum temperatures? Or averages?

Response: We have added the information in line 145.

L113: statistical tests used and test statistics should be reported here and throughout. For example, how were these confidence intervals generated? In addition to potentially conflated factors (e.g., spatial or temporal replication), it appears that at least some of the variable (e.g., flight #) are not normally distributed, so it is unclear whether, for example, confidence intervals generated assuming a normal distribution would be appropriate here.

Response: We now applied GLMM for statistical analysis and accounted spatial (site effect) and temporal (study day) by adding them as a random factor in our analysis. DurgaDiff does not rely on normally distributed data and estimates effect size by bootstrapping data (we applied bootstrapping 1000 times in our analysis). Details of statistical analyses are provided in method (line 131-141), and in result (line 146-158).

L126: remove 'noticeable'

Response: We remove the word from the sentence in line 162.

L138: 'insects' should be singular

Response: We corrected the word in line 174.

L152: Change 'denoted' to 'demonstrated' or equivalent?

Response: We changed the word to demonstrated in line 188.

L159-162: interested observation, I would suggest expanding on the details of this observation a bit.

Response: We have expanded the information in line 196-201.

L162: '... and may also contributed' needs to be reworded

Response: We have reworded the sentence in line 200.

L175: 'significance' should be 'significant'

Response: We have corrected the word in sentence in line 214.

L185: Word missing in this sentence, maybe 'are experiencing'?

Response: We have corrected the word in line 230.

Fig 1: I would suggest either plotting kernel density estimates or boxplots, but not both - I find this distracting (especially given that the raw data is also plotted)

Response: Thank you for the suggestion. We have re-created the figure 1 using boxplots in line 240.

Resubmission

First decision letter

MS ID#: bio.062091

MS Title: Heatwaves reduce mating frequency in an aquatic insect

Authors: Md Tangigul Haque; Shatabdi Paul; kawsar khan

MS ID#: bio.062091

MS Title: Heatwaves reduce mating frequency in an aquatic insect

Authors: Md Tangigul Haque; Shatabdi Paul; kawsar khan

Dear Dr Khan,
I have now reached a decision on the above manuscript.

The reviewer reports are shown at the bottom of this email or can be accessed, together with a copy of this decision letter, by going to:

As you will see, the reviewers raised a number of substantial criticisms that prevent me from accepting the paper at this stage. They suggest, however, that a revised version might prove acceptable, if you can address their concerns. In particular, there are still major issues with the statistical analysis and the interpretation of the results. If you think that you can deal satisfactorily with the criticisms on revision, I would be pleased to see a revised manuscript. We would then return it to the reviewers.

At this stage, we also ask you to ensure your manuscript complies with our formatting guidelines. Provided you are able to fully address the referees' comments, we are positive about publication of your paper (we accept over 95% of revision submissions) and therefore hope you won't mind any extra work involved in reformatting your manuscript at this point.

Please ensure that you clearly highlight all changes made in the revised manuscript. Please avoid using 'Tracked changes' in Word files as these are lost in PDF conversion.

I should be grateful if you would also provide a point-by-point response detailing how you have dealt with the points raised by the reviewers in the 'Response to Reviewers' box. Please attend to all of the reviewers' comments. If you do not agree with any of their criticisms or suggestions please explain clearly why this is so.

Reviewer 1

Comments for the author

The authors have responded to the first-round reviews, adding clearer descriptions of their heat-wave definition (although see below), expanding the statistical-methods section to include GLMMs, and providing more context for sampling design and weather measurements. These additions substantially improve methodological transparency. Overall, the key novel finding of the paper is the effect of

However, some significant issues remain that need to be addressed to strengthen the manuscript:

First, the definition of 'heat waves' is still not entirely clear, and needs further explanation (see specific notes below)

Second, while further details are now included on the observation structures, the temporal structure and spacing of these observations is not clear. E.g., there are three observation periods in each heat wave conditions across three months, but were all the heat wave days in one of those months, and 'normal' days in another? Without knowing how these are structured (especially since they aren't experimentally randomized), it is not possible to clearly disentangle any potential effects of e.g., seasonal shifts in mating frequency.

Third, there are still some aspects of the statistical methods and reporting that require some additional clarification and edits. This includes (a) the use of random effects in specific models (see specific comments below), (b) the lack of reporting of some significance (e.g., lines 150-152), and (c) the use of both generalized linear mixed models, but also reporting of confidence intervals (from the Durga package) where it is not clear if they are from modeled outputs from the statistical models presented (and thus misleading).

Finally, throughout, some results are reported as 'marginally' significant, even at p values above 0.1. While I'm supportive of graded interpretation of probability broadly (e.g., that don't interpret

p-values of .049 and .051 very differently), these p-values really are not supportive of any level of significance, and should be interpreted accordingly.

Specific comments

There are numerous grammatical and typographical errors throughout the manuscript. I have included some of these below, but a close language edit will be important.

L25: 'Ectotherms, particularly insects...' this sentence is a bit vague, edit to be more specific.

L31: Given the timescales being investigated here (changes over days), it is unlikely that the changes observed here are associated with changes in the populations, so would suggest editing to something more like 'activity' to clarify.

L36-38: Awkward sentence - edit for clarity

L52-53: this definition is still not entirely clear to me, and in particular I think a threshold is required for this definition. E.g., is three days 0.1C above average a heat wave? Or does the difference need be higher? Defined as peak temperature, or sustained high temperatures? It is unclear here what definition the authors actually used which is central to their findings.

L59-62: cite examples for this?

L86: perhaps 'temporal' should be 'temperate'?

L93-94: Ditto above on defining heat wave days. Also was there a single, separate heatwave days recorded in each of these months? Or are 'heatwave days', for example, clustered within one of those months? This is important for decoupling phenology and environmental conditions.

L102-103: "proxy of damselflies foraging behaviour" should be "proxy for damselflies' foraging behaviour".

L107 "recorded data of the total number of flights an individual" should be "...flights by an individual...".

L109: "spent 45 minutes to record flight number" - 'record' should be 'recording'

L134-`137: For the mating and abundance models, it seems as though observation day was not included as a random effect? Not clear why this is different than e.g. the flight number model, and this could be an important source of non-independence in data.

L138-141: Does DurgaDiff operate on the 'raw' data, or take estimates from a modeled output (e.g., incorporating random effects from the glmmTMB)? It would seem like the former, in which case it is a bit misleading to plot confidence intervals from this approach that does not reflect the statistical models used.

L145-146: If I'm interpreting correctly, this is three measurements for each condition? If this is case, I would report raw values (standard deviation estimates are not particularly meaningful for this # of observations)

L147: I don't think that non-significant results should be reported as 'marginally higher', this is misleading (this also applies L161-162 and other places in the manuscript).

L150-152: not clear if there is statistical support for this result

L178-179: this is referring to a non-significant results, so edit to reflect that more clearly

L183: Unclear what results this is referring to, or support for them

L191-192: '...exhibit microhabitat variation across developmental stage'... stage should 'stages'

L196-197: this observation should be explained a bit more - was this specifically during heat waves, for example?

L203-205: the connections to phenology (which would presumably occur on different timescales?) need to be clarified, or this sentence removed.

Fig 1: If the 'DurgaPlots' are showing confidence intervals calculated directly from the data, rather than a modeled output (e.g., that includes random effects, etc), I think this is a bit misleading and could lead to incorrect interpretations about the significance of comparisons.

Reviewer 2

Comments for the author

I previously reviewed an earlier version of this manuscript, and I acknowledge that the authors have made substantial improvements in response to the initial comments. I appreciate their careful attention to the suggested revisions. However, I still find significant flaws in the data analysis that, in my opinion, compromise the validity of some key findings.

My major concerns are as follows:

1) In the revised manuscript, the authors have appropriately employed Generalized Linear Mixed Models (GLMMs) for each response variable (flight activity, abundance, and mating activity). However, for flight number, the rationale for excluding Sex as a fixed factor is not provided. This omission is particularly relevant, as the manuscript repeatedly states that "flight number per minute was higher for males than for females" (lines 151-152). This interpretation also appears in the abstract (lines 31-32) and the discussion (lines 183-185), yet these sex differences were not formally (statistically) tested. Including Sex as a fixed factor would provide a more robust basis for these interpretations.

2) Regarding the analysis of flight number, the manuscript states that "during the heatwave, flight number was *marginally* higher in males and females..." (lines 146-150). However, this is misleading. Both the GLMM and the DurgaDiff analyses show no statistically significant differences between normal and heatwave days, implying that the observed variation is not meaningful. This point should be corrected not only in the main text but also in the abstract (line 30) and in the legend of Figure 1 (line 240).

3) It would be important to include detailed tables showing the output of the GLMMs for each response variable. Presenting the results only in the text makes them difficult to follow, especially since the outputs of both the GLMM and the DurgaDiff analyses are often combined within the same parentheses, which adds confusion.

4) In the Introduction, the definition of heatwaves involves increased day and nighttime temperatures relative to a climate reference value over three consecutive days. However, the Methods section (line 93) does not specify what that reference value is. In the previous version of the manuscript, a threshold of 35°C was mentioned, but this detail is now missing. Please clarify this critical methodological point.

In summary, while the research question remains relevant and timely, the manuscript still presents substantial limitations—particularly in its statistical analyses—that need to be addressed.

Reviewer's Responses to Questions

Experimental quality

Does each figure have the proper controls?

If 'No', please indicate reasons in Comments for Author box below.

Reviewer #1:

- Yes

Reviewer #2:

- Yes

Were the data analyzed using appropriate statistical tests?

If 'No', please indicate reasons in Comments for Author box below.

Reviewer #1:

- Yes

Reviewer #2:

- No

Reproducibility

Were experiments performed using adequate number of biological replicates?

If 'No', please indicate reasons in Comments for Author box below.

Reviewer #1:

- No

Reviewer #2:

- Yes

Does the methods section provide sufficient detail to permit reproducibility?

If 'No', please indicate reasons in Comments for Author box below.

Reviewer #1:

- Yes

Reviewer #2:

- Yes

Completeness

Are the manuscript's conclusions supported by the data?

If 'No', please indicate reasons in Comments for Author box below.

Reviewer #1:

- Yes

Reviewer #2:

- No

Scholarship

Do the authors cite and discuss the merits of data that would argue for and against their conclusion?

If 'No', please indicate reasons in Comments for Author box below.

Reviewer #1:

- Yes

Reviewer #2:

- Yes

Does the manuscript title & abstract accurately reflect the contents of the manuscript, without hyperbole?

If 'No', please indicate reasons in Comments for Author box below.

Reviewer #1:

- Yes

Reviewer #2:

- Yes

First revision

Author response to reviewers' comments

Response to the reviewers and editors

Dear editor,

Thank you for your valuable suggestions and giving us the opportunity to improve the manuscript again. We addressed all the comments that reviewers suggested and incorporated all changes in our main article with line numbers for your convenience. In addition, we have formatted our manuscript according to the journal guidelines. Please find our responses below:

Reviewer 1:

First, the definition of 'heat waves' is still not entirely clear, and needs further explanation (see specific notes below)

Response: Thank you for pointing this out. We defined the term 'heatwaves' into a more simplified form in the introduction (lines 60 -63) and more detailed information in the methods (lines 207-214 and 219-220).

Second, while further details are now included on the observation structures, the temporal structure and spacing of these observations is not clear. E.g., there are three observation periods in each heat wave conditions across three months, but were all the heat wave days in one of those months, and 'normal' days in another? Without knowing how these are structured (especially since they aren't experimentally randomized), it is not possible to clearly disentangle any potential effects of e.g., seasonal shifts in mating frequency.

Response: We collected data on normal days an average of 9.6 ± 1.5 SE days after the heatwave sampling periods. We added this information in the lines 218-219.

Third, there are still some aspects of the statistical methods and reporting that require some additional clarification and edits. This includes (a) the use of random effects in specific models (see

specific comments below), (b) the lack of reporting of some significance (e.g., lines 150-152), and (c) the use of both generalized linear mixed models, but also reporting of confidence intervals (from the Durga package) where it is not clear if they are from modeled outputs from the statistical models presented (and thus misleading).

Response: As per reviewer's suggestion, we now removed the DurgaDiff function from our analyses as DurgaDiff cannot report confidence interval from the model. We now statistically tested whether the effects of heatwaves differed between sexes and reported the analysis in the lines 111-113. We also included observation day as a random factor and modified our models in lines 260-263, however, it did not change our results.

Finally, throughout, some results are reported as 'marginally' significant, even at p values above 0.1. While I'm supportive of graded interpretation of probability broadly (e.g., that don't interpret p-values of .049 and .051 very differently), these p values really are not supportive of any level of significance, and should be interpreted accordingly.

Response: According to reviewers suggestion, we now removed marginally higher flight number, and reported our results as not significant throughout the manuscript in lines 107-110, and lines 122-123.

Specific comments

L25: 'Ectotherms, particularly insects...' this sentence is a bit vague, edit to be more specific.

Response: We clarified the sentence in lines 25-26.

L31: Given the timescales being investigated here (changes over days), it is unlikely that the changes observed here are associated with changes in the populations, so would suggest editing to something more like 'activity' to clarify.

Response: We changed the word mating activity to more specific mating frequency in lines 30-31.

L36-38: Awkward sentence - edit for clarity

Response: We clarified the sentence in lines 35-37.

L52-53: this definition is still not entirely clear to me, and in particular I think a threshold is required for this definition. E.g., is three days 0.1C above average a heat wave? Or does the difference need be higher? Defined as peak temperature, or sustained high temperatures? It is unclear here what definition the authors actually used which is central to their findings.

Response: We clarified the definition again in lines 60-63. According to Bureau of Meteorology Australia, there is no specific threshold for heatwave definition. However, in this study we applied a threshold of at least 4°C above the historical average to define a heatwave. We added details information in the method section in lines 207-214.

L59-62: cite examples for this?

Response: We added the reference for this information in lines 72-73. L86: perhaps 'temporal' should be 'temperate'?

Response: We corrected the word in line 101.

L93-94: Ditto above on defining heat wave days. Also was there a single, separate heatwave days recorded in each of these months? Or are 'heatwave days', for example, clustered within one of those months? This is important for decoupling phenology and environmental conditions.

Response: We collected data on normal days an average of 9.6 ± 1.5 SE days after the heatwaves sampling periods. We added this information in the lines 218-219.

L102-103: "proxy of damselflies foraging behaviour" should be "proxy for damselflies' foraging behaviour".

Response: We changed the word in line 224.

L107 "recorded data of the total number of flights an individual" should be "...flights by an individual...".

Response: We changed the word in line 229.

L109: "spent 45 minutes to record flight number" - 'record' should be 'recording'

Response: We changed the word in line 231.

L134-137: For the mating and abundance models, it seems as though observation day was not included as a random effect? Not clear why this is different than e.g. the flight number model, and this could be an important source of nonindependence in data.

Response: Now we added observation day as a random factor for both mating and abundance models. We reported that the results were exactly same with previous results in lines 260-263.

L138-141: Does DurgaDiff operate on the 'raw' data, or take estimates from a modeled output (e.g., incorporating random effects from the glmmTMB)? It would seem like the former, in which case it is a bit misleading to plot confidence intervals from this approach that does not reflect the statistical models used.

Response: DurgaDiff uses raw data to measure the confidence interval and estimate mean differences. Hence, we removed the DurgaDiff function from our analyses for clarity.

L145-146: If I'm interpreting correctly, this is three measurements for each condition? If this is case, I would report raw values (standard deviation estimates are not particularly meaningful for this # of observations)

Response: We revised the sentence and reported the raw values for each condition in lines 106-107.

L147: I don't think that non-significant results should be reported as 'marginally higher', this is misleading (this also applies L161-162 and other places in the manuscript).

Response: We now removed the word marginally higher and directly reported our results as non-significant result throughout the manuscript in lines 122-123, and throughout the manuscript.

L150-152: not clear if there is statistical support for this result

Response: We agreed with the reviewer and tested statistically. However, we did not find support for the result, hence we added the information as non-significant in lines 111-113.

L178-179: this is referring to a non-significant results, so edit to reflect that more clearly

Response: We adjusted the sentence in lines 140-141.

L183: Unclear what results this is referring to, or support for them

Response: We adjusted our results and removed the marginally higher flight number from the findings as it was not-significant, we re-wrote the paragraph and removed the line from the paragraph. We added the updated information in lines 140-145.

L191-192: '...exhibit microhabitat variation across developmental stage'... stage should 'stages'

Response: We corrected the word in line 157.

L196-197: this observation should be explained a bit more - was this specifically during heat waves, for example?

Response: We now added clear explanation in lines 161-163.

L203-205: the connections to phenology (which would presumably occur on different timescales?) need to be clarified, or this sentence removed.

Response: For clarity, we removed the sentence from the manuscript.

Fig 1: If the 'DurgaPlots' are showing confidence intervals calculated directly from the data, rather than a modeled output (e.g., that includes random effects, etc), I think this is a bit misleading and could lead to incorrect interpretations about the significance of comparisons.

Response: DurgaDiff uses raw data to measure the confidence interval and estimate mean differences. Hence, we removed the DurgaDiff function from our analyses for clarity. Now we added boxplots to show the relationship of flight number, abundance, and mating number between heatwaves and normal days in figure 1 and 2.

Reviewer 2:

1) In the revised manuscript, the authors have appropriately employed Generalized Linear Mixed Models (GLMMs) for each response variable (flight activity, abundance, and mating activity). However, for flight number, the rationale for excluding Sex as a fixed factor is not provided. This omission is particularly relevant, as the manuscript repeatedly states that "flight number per minute was higher for males than for females" (lines 151-152). This interpretation also appears in the abstract (lines 31-32) and the discussion (lines 183-185), yet these sex differences were not formally (statistically) tested. Including Sex as a fixed factor would provide a more robust basis for these interpretations.

Response: Thank you for your suggestions. We added an additional model including sex as a fixed factor to identify a robust interaction between days (heatwaves and normal days) and sex. We removed the previous line and added the output from the statistical result, suggesting that the effect of heatwaves did not differ significantly between sexes in lines 111-113. We also removed the line from the abstract and discussion section as we adjusted our results, and according to the first reviewers' suggestions.

2) Regarding the analysis of flight number, the manuscript states that "during the heatwave, flight number was *marginally* higher in males and females..." (lines 146-150). However, this is misleading. Both the GLMM and the DurgaDiff analyses show no statistically significant differences between normal and heatwave days, implying that the observed variation is not meaningful. This point should be corrected not only in the main text but also in the abstract (line 30) and in the legend of Figure 1 (line 240).

Response: We removed DurgaDiff from our analyses as it cannot use model output for determining confidence interval. Now, throughout the manuscript we mentioned the observation output as not significant and removed marginally higher from sentences including abstract and in the legend of Figure 1 (lines 403-404).

3) It would be important to include detailed tables showing the output of the GLMMs for each response variable. Presenting the results only in the text makes them difficult to follow, especially since the outputs of both the GLMM and the DurgaDiff analyses are often combined within the same parentheses, which adds confusion.

Response: For clarity, we removed the DurgaDiff function from our analyses and used only GLMMs for observations. In addition, we added a table (Table 1) describing all the models and respective

outputs of the model in lines 436-439.

4) In the Introduction, the definition of heatwaves involves increased day and nighttime temperatures relative to a climate reference value over three consecutive days. However, the Methods section (line 93) does not specify what that reference value is. In the previous version of the manuscript, a threshold of 35°C was mentioned, but this detail is now missing. Please clarify this critical methodological point.

Response: We adjusted the heatwaves definition for clarity in our study. We defined heatwaves as a period when three consecutive days average temperatures substantially exceed the historical monthly average temperatures for the respective months, based on 1859-2020 climate record. All heatwave days data were collected when the temperature was above 35°C. However, as suggested by the reviewer, temperature alone is not a good definition of a heatwave, as it depends on the local climate and its comparison with historical data. Now, we applied a threshold of at least 4°C above the historical average to define a heatwave which better reflect the terminology of heatwave and provide easier option to compare across geographical regions. We have clarified the details information in the introduction section in lines 60-63 and methods section in lines 207-214 and 219-220.

Second decision letter

MS ID#: bio.062091R1

MS Title: Heatwaves reduce mating frequency in an aquatic insect

Authors: Md Tangigul Haque; Shatabdi Paul; kawsar khan

I am happy to tell you that your manuscript has been accepted for publication in Biology Open, pending our standard publication integrity checks. It was accepted on 07 July 2025.